# Knowledge Distillation from A Stronger Teacher

**Tao Huang**[1,2]  **Shan You**[1]*  **Fei Wang**[3]  **Chen Qian**[1]  **Chang Xu**[2]

[1]SenseTime Research
[2]School of Computer Science, Faculty of Engineering, The University of Sydney
[3]University of Science and Technology of China

## Abstract

Unlike existing knowledge distillation methods focus on the baseline settings, where the teacher models and training strategies are not that strong and competing as state-of-the-art approaches, this paper presents a method dubbed DIST to distill better from a stronger teacher. We empirically find that the discrepancy of predictions between the student and a stronger teacher may tend to be fairly severer. As a result, the exact match of predictions in KL divergence would disturb the training and make existing methods perform poorly. In this paper, we show that simply preserving the relations between the predictions of teacher and student would suffice, and propose a correlation-based loss to capture the intrinsic inter-class relations from the teacher explicitly. Besides, considering that different instances have different semantic similarities to each class, we also extend this relational match to the intra-class level. Our method is simple yet practical, and extensive experiments demonstrate that it adapts well to various architectures, model sizes and training strategies, and can achieve state-of-the-art performance consistently on image classification, object detection, and semantic segmentation tasks. Code is available at: https://github.com/hunto/DIST_KD.

## 1   Introduction

The advent of automatic feature engineering fuels deep neural networks to achieve remarkable success in a plethora of computer vision tasks, such as image classification [17, 19, 38, 48, 53], object detection [2, 23], and semantic segmentation [5, 54]. In the path of pursuing better performance, current deep learning models generally grow deeper and wider [13, 45]. However, such heavy models are clumsy to deploy in practice due to the limitations of computational and memory resources. For an efficient model with competitive performance to those larger models, knowledge distillation (KD) [16] has been proposed to boost the performance of the efficient model (student) by distilling the knowledge of a larger model (teacher) during training.

The essence of knowledge distillation relies on how to formulate and transfer the knowledge from teacher to student. The most intuitive yet effective approach is to match the probabilistic prediction (response) scores between the teacher and student via Kullback–Leibler (KL) divergence [16]. In this way, the student can be guided with more informative signals during training, and is thus expected to have more promising performance than that being trained stand-alone. Besides this vanilla prediction match, other works [11, 14, 34, 41] also investigate the knowledge within intermediate representations to further boost the distillation performance, but this usually induces additional training cost as a consequence. For example, OFD [14] proposes to distill the information via multiple intermediate layers, but requires additional convolutions for feature alignments; CRD [41] introduces a contrastive loss to transfer pair-wise relationships, but it needs to hold a memory bank for all 128-d features of ImageNet images, and produces additional 260M FLOPs of computation cost.

---

*Correspondence to: Shan You <youshan@sensetime.com>.

36th Conference on Neural Information Processing Systems (NeurIPS 2022).

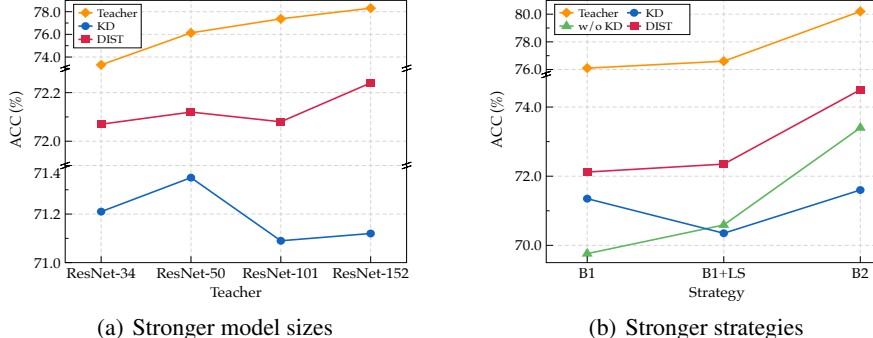

(a) Stronger model sizes          (b) Stronger strategies

Figure 1: Comparisons of KD and our proposed DIST on ImageNet with different teachers. (a) The ResNet-18 students are trained using baseline strategy with different model sizes of the teacher. (b) The ResNet-18 students are trained using different strategies with ResNet-50 teachers.

Recently, a few studies [8, 29, 39] have been performed to address the poor learning issue of the student network when the student and teacher model sizes significantly differ. For example, TAKD [29] proposes to reduce the discrepancy of teacher and student by resorting to an additional teaching assistant of moderate model size; DGKD [39] further improves TAKD by densely gathering all the assistant models to guide the student. However, increasing the model size is only one of the popular approaches to have a stronger teacher. There lacks a thorough analysis on the training strategies to derive a stronger teacher and their effect on KD. Most importantly, a generic enough solution is preferred to address the difficulty of KD brought by stronger teachers, rather than struggling to deal with different types of stronger teachers (with larger model size or stronger training strategy) individually.

To understand what makes a stronger teacher and their effect on KD, we systematically study the prevalent strategies for designing and training deep neural networks, and show that:

- Beyond scaling up the model size, a stronger teacher can also be derived through advanced training strategies, *e.g.*, label smoothing and data augmentation [51]. However, given a stronger teacher, the student's performance on the vanilla KD could be dropped, even worse than training from scratch without KD, as shown in Figure 1.

- The discrepancy between teacher and student tends to get fairly larger when we switch their training strategy to a stronger one (see Figure 2). In this case, an exact recovery of predictions via KL divergence could be challenging and lead to the failure of vanilla KD.

- Preserving the *relation of predictions* between teacher and student is sufficient and effective. When transferring the knowledge from teacher to student, what we really care about is preserving the preference (relative ranks of predictions) by the teacher, instead of recovering the absolute values accurately. Correlation between teacher and student predictions could be favored to relax the exact match of KL divergence and distill the intrinsic relations.

In this paper, we thus leverage the Pearson correlation coefficient [33] as a new match manner to replace the KL divergence. In addition, besides the *inter-class relations* in prediction vector (see Figure 3), with the intuition that different instances have different spectrum of similarities with respect to each class, we also propose to distill the *intra-class relations* for further boosting the performance as Figure 3. Concretely, for each class, we gather its corresponding predicted probabilities of all instances in a batch, then transfer this relation from teacher to student. Our proposed method (dubbed DIST) is super simple, efficient, and practical, which can be implemented with only several lines of code (see Appendix A.1) and has almost the same training cost as the vanilla KD. As a result, the student can be liberated from the burden of matching the exact output of a strong teacher, but only be guided appropriately to distill those truly informative relations.

Extensive experiments are conducted on benchmark datasets to verify our effectiveness on various tasks, including image classification, object detection, and semantic segmentation. Experimental results show that our DIST significantly outperforms vanilla KD and those sophisticatedly-designed state-of-the-art KD methods. For example, with the same baseline settings on ImageNet, our DIST

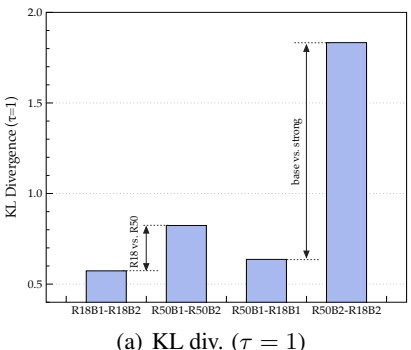
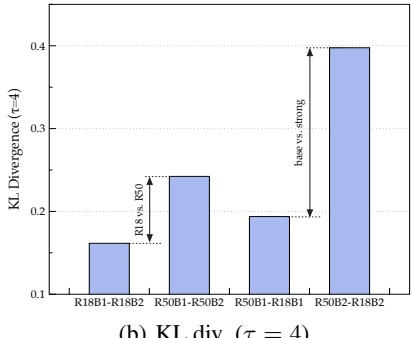

| (a) KL div. ($\tau = 1$) | (b) KL div. ($\tau = 4$) |
|---|---|

Figure 2: Discrepancy between the predictions of models trained standalone with different strategies on ImageNet validation set. *R18B1* represents ResNet-18 trained with strategy B1 for instance. Details of training strategies B1 and B2 refer to Table 1.

achieves the highest 72.07% accuracy on ResNet-18. With the stronger strategy, our method obtains 82.3% accuracy on the recent transformer Swin-T [27], improving KD by 1%.

## 2 Revisiting Prediction Match of KD

In vanilla knowledge distillation [16], the knowledge is transferred from a pre-trained teacher model to a student model by minimizing the discrepancy between the prediction scores of the teacher and student models.

Formally, with the logits $\boldsymbol{Z}^{(\text{s})} \in \mathbb{R}^{B \times C}$ and $\boldsymbol{Z}^{(\text{t})} \in \mathbb{R}^{B \times C}$ of student and teacher networks, where $B$ and $C$ denote batch size and the number of classes, respectively, the vanilla KD loss [16] is represented as

$$\mathcal{L}_{\text{KD}} := \frac{\tau^2}{B} \sum_{i=1}^{B} \text{KL}(\boldsymbol{Y}_{i,:}^{(\text{t})}, \boldsymbol{Y}_{i,:}^{(\text{s})}) = \frac{\tau^2}{B} \sum_{i=1}^{B} \sum_{j=1}^{C} Y_{i,j}^{(\text{t})} \log \left( \frac{Y_{i,j}^{(\text{t})}}{Y_{i,j}^{(\text{s})}} \right), \tag{1}$$

where KL refers to Kullback–Leibler divergence with

$$\boldsymbol{Y}_{i,:}^{(\text{s})} = softmax(\boldsymbol{Z}_{i,:}^{(\text{s})}/\tau), \quad \boldsymbol{Y}_{i,:}^{(\text{t})} = softmax(\boldsymbol{Z}_{i,:}^{(\text{t})}/\tau), \tag{2}$$

being the probabilistic prediction vectors, and $\tau$ is the temperature factor to control the softness of logits.

In addition to the teacher's soft targets in Eq.(1), KD [16] stated that it is beneficial to train the student together with ground-truth labels, and the overall training loss is composed of the original classification loss $\mathcal{L}_{\text{cls}}$ and KD loss $\mathcal{L}_{\text{KD}}$, *i.e.*,

$$\mathcal{L}_{\text{tr}} = \alpha \mathcal{L}_{\text{cls}} + \beta \mathcal{L}_{\text{KD}}, \tag{3}$$

where $\mathcal{L}_{\text{cls}}$ is usually the cross-entropy loss between the predictions of student network and ground-truth labels, $\alpha$ and $\beta$ are factors for balancing the losses.

### 2.1 Catastrophic discrepancy with a stronger teacher

As illustrated in Section 1, the effect of a teacher on KD has not been sufficiently investigated, especially when the performance of pre-trained teacher grows stronger, such as with larger model size or being trained with more advanced and competing strategies, *e.g.*, label smoothing, mix-up [51], auto augmentations [9], *etc*. With this regard, as Figure 2, we train ResNet-18 and ResNet-50 standalone with strategy B1 and strategy B2[2], and obtain 4 trained models (*R18B1*, *R18B2*, *R50B1*, and *R50B2* with accuracies 69.76%, 73.4%, 76.13%, and 78.5%, respectively), then compare their discrepancy using KL divergence ($\tau = 1$ and $\tau = 4$) on the predicted probabilities $\boldsymbol{Y}$. We have the following observations:

---

[2]Training with B2 obtains higher accuracy compared to B1, *e.g.*, 73.4% (B2) vs. 69.8% (B1) on ResNet-18.

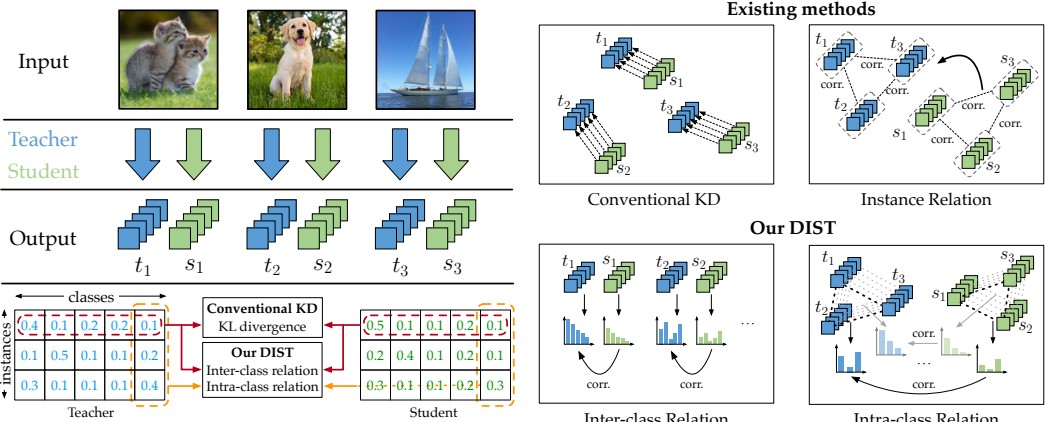

Figure 3: **Difference between our DIST and existing KD methods.** Conventional KD matches the outputs of student ($s \in \mathbb{R}^5$) to teacher ($t \in \mathbb{R}^5$) point-wisely; instance relation methods operate on the feature level and measure the internal correlations (corr.) between instances in student and teacher separately, then transfer the teacher's correlations to student. Our DIST proposes to maintain the inter-class and intra-class relations between student and teacher. Inter-class relation: correlation between the predicted probabilistic distributions on each instance of teacher and student. Intra-class relation: correlation of the probabilities of all the instances on each class.

- The outputs of ResNet-18 do not change much with the stronger strategy compared to ResNet-50. This implies that the representational capacity limits the student's performance, and it tends to be fairly challenging for the student to exactly match the teacher's outputs as their discrepancy becomes larger.

- When the teacher and student models are trained with a stronger strategy, the discrepancy between teacher and student would be larger. This indicates that when we adopt KD with a stronger training strategy, the misalignment between KD loss and classification loss would be severer, thus disturbing the student's training.

As a result, the exact match (*i.e.*, the loss reaches the minimal if and only if the teacher and student outputs are exactly identical) with KL divergence seems way too overambitious and demanding since the discrepancy between student and teacher can be considerably huge. Since the exact match can be detrimental with a stronger teacher, our intuition is to develop a relaxed manner for matching the predictions between the teacher and student.

## 3 DIST: Distillation from A Stronger Teacher

### 3.1 Relaxed match with relations

The prediction scores indicate the teacher's confidence (or preference) over all classes. For a relaxed match of predictions between the teacher and student, we are motivated to consider what we really care about for the teacher's output. Instead of the exact probabilistic values, actually, during inference, we are only concerned about their **relations**, *i.e.*, relative ranks of predictions of teacher.

In this way, for some metric $d(\cdot, \cdot)$ with $\mathbb{R}^C \times \mathbb{R}^C \to \mathbb{R}^+$, the exact match can be formulated that $d(\boldsymbol{a}, \boldsymbol{b}) = 0$ if $\boldsymbol{a} = \boldsymbol{b}$ for any two prediction vector as $\boldsymbol{Y}_{i,:}^{(\text{s})}$ and $\boldsymbol{Y}_{i,:}^{(\text{t})}$ in the KL divergence of Eq.(1). Then as a relaxed match, we can introduce additional mappings $\phi(\cdot)$ and $\psi(\cdot)$ with $\mathbb{R}^C \to \mathbb{R}^C$ such that

$$d(\phi(\boldsymbol{a}), \psi(\boldsymbol{b})) = d(\boldsymbol{a}, \boldsymbol{b}), \forall \boldsymbol{a}, \boldsymbol{b} \tag{4}$$

Therefore, $d(\boldsymbol{a}, \boldsymbol{b}) = 0$ does not necessarily require $\boldsymbol{a}$ and $\boldsymbol{b}$ should be exactly the same. Nevertheless, since we care about the relation within $\boldsymbol{a}$ or $\boldsymbol{b}$, the mappings $\phi$ and $\psi$ should be isotone and do not affect the semantic information and inference result of the prediction vector.

With this regard, a simple yet effective choice for the isotone mapping is the positive linear transformation, namely,

$$d(m_1\boldsymbol{a} + n_1, m_2\boldsymbol{b} + n_2) = d(\boldsymbol{a}, \boldsymbol{b}), \tag{5}$$

where $m_1$, $m_2$, $n_1$, and $n_2$ are constants with $m_1 \times m_2 > 0$. As a result, this match could be invariant under separate changes in scale and shift for the predictions. Actually, to satisfy the property Eq.(5), we can thus adopt the widely-used Pearson's distance as the metric, *i.e.*,

$$d_{\mathrm{p}}(\boldsymbol{u}, \boldsymbol{v}) := 1 - \rho_{\mathrm{p}}(\boldsymbol{u}, \boldsymbol{v}). \tag{6}$$

$\rho_{\mathrm{p}}(\boldsymbol{u}, \boldsymbol{v})$ is the Pearson correlation coefficient between two random variables $\boldsymbol{u}$ and $\boldsymbol{v}$,

$$\rho_{\mathrm{p}}(\boldsymbol{u}, \boldsymbol{v}) := \frac{\mathrm{Cov}(\boldsymbol{u}, \boldsymbol{v})}{\mathrm{Std}(\boldsymbol{u})\mathrm{Std}(\boldsymbol{v})} = \frac{\sum_{i=1}^{C}(u_i - \bar{u})(v_i - \bar{v})}{\sqrt{\sum_{i=1}^{C}(u_i - \bar{u})^2 \sum_{i=1}^{C}(v_i - \bar{v})^2}} \tag{7}$$

where $\mathrm{Cov}(\boldsymbol{u}, \boldsymbol{v})$ is the covariance of $\boldsymbol{u}$ and $\boldsymbol{v}$, $\bar{u}$ and $\mathrm{Std}(\boldsymbol{u})$ denote the mean and standard derivation of $\boldsymbol{u}$, respectively.

In this way, we can define the **relation as correlation**. More specifically, and the original exact match in vanilla KD [16] can thus be relaxed and replaced by maximizing the linear correlation to preserve the relation of teacher and student on the probabilistic distribution of each instance, which we call *inter-class relation*. Formally, for each pair of prediction vector $\boldsymbol{Y}_{i,:}^{(\mathrm{s})}$ and $\boldsymbol{Y}_{i,:}^{(\mathrm{t})}$, the inter-relation loss can be formulated as

$$\mathcal{L}_{\mathrm{inter}} := \frac{1}{B} \sum_{i=1}^{B} d_{\mathrm{p}}(\boldsymbol{Y}_{i,:}^{(\mathrm{s})}, \boldsymbol{Y}_{i,:}^{(\mathrm{t})}). \tag{8}$$

Some isotone mappings or metrics can also be used to relax the match as Eq.(4), such as cosine similarity investigated empirically in Section 4.5; other more advanced and delicate choices could be left as future work.

## 3.2 Better distillation with intra-relations

Besides the inter-class relation, where we transfer the relation of multiple classes in each instance, the prediction scores of multiple instances in each class are also informative and useful. This scores indicate the similarities of multiple instances to one class. For instance, suppose we have three images containing "cat", "dog", and "plane", respectively, and they have three prediction scores on the 'cat' class, denoted as $e$, $f$, and $g$. Generally, the picture "cat" should have the largest score to the "cat" class, while the "plane" should have the smallest score since it is inanimate. This relation of "$e > f > g$" could also be transferred to the student. Besides, even for the images from the same class, the intrinsic intra-class variance of the semantic similarities is actually also informative. It indicates the prior from the teacher that *which one is more reliable to cast in this class*.

Therefore, we also encourage to distill this *intra-relation* for better performance. Actually, define prediction matrix $\boldsymbol{Y}^{(\mathrm{s})}$ and $\boldsymbol{Y}^{(\mathrm{t})}$ with each row as $\boldsymbol{Y}_{i,:}^{(\mathrm{s})}$ and $\boldsymbol{Y}_{i,:}^{(\mathrm{t})}$, then the above inter-relation is to maximize the correlation row-wisely (see Figure 3). In contrast, for intra-relation, the corresponding loss is thus to maximize the correlation column-wisely, *i.e.*,

$$\mathcal{L}_{\mathrm{intra}} := \frac{1}{C} \sum_{j=1}^{C} d_{\mathrm{p}}(\boldsymbol{Y}_{:,j}^{(\mathrm{s})}, \boldsymbol{Y}_{:,j}^{(\mathrm{t})}). \tag{9}$$

As a result, the overall training loss $\mathcal{L}_{\mathrm{tr}}$ can be composed of the classification loss, inter-class KD loss, and intra-class KD loss, *i.e.*,

$$\mathcal{L}_{\mathrm{tr}} = \alpha \mathcal{L}_{\mathrm{cls}} + \beta \mathcal{L}_{\mathrm{inter}} + \gamma \mathcal{L}_{\mathrm{intra}}, \tag{10}$$

where $\alpha$, $\beta$, and $\gamma$ are factors for balancing the losses. In this way, via the relation loss, we have endowed the student with freedom more or less to match the teacher network's output adaptively, thus boosting the distillation performance to a great extent.

Table 1: **Training strategies on image classification tasks.** *BS*: batch size; *LR*: learning rate; *WD*: weight decay; *LS*: label smoothing; *EMA*: model exponential moving average; *RA*: RandAugment [9]; *RE*: random erasing; *CJ*: color jitter.

| Strategy | Dataset | Epochs | Total BS | Initial LR | Optimizer | WD | LS | EMA | LR scheduler | Data augmentation |
|---|---|---|---|---|---|---|---|---|---|---|
| A1 | CIFAR-100 | 240 | 64 | 0.05 | SGD | $5 \times 10^{-4}$ | - | - | $\times 0.1$ at 150,180,210 epochs | crop + flip |
| B1 | ImageNet | 100 | 256 | 0.1 | SGD | $1 \times 10^{-4}$ | - | - | $\times 0.1$ every 30 epochs | crop + flip |
| B2 | ImageNet | 450 | 768 | 0.048 | RMSProp | $1 \times 10^{-5}$ | 0.1 | 0.9999 | $\times 0.97$ every 2.4 epochs | {*B1*} + RA + RE |
| B3 | ImageNet | 300 | 1024 | 5e-4 | AdamW | $5 \times 10^{-2}$ | 0.1 | - | cosine | {*B2*} + CJ + Mixup + CutMix |

Table 2: **Evaluation results of baseline settings on ImageNet.** We use ResNet-34 and ResNet-50 released by Torchvision [28] as our teacher networks, and follow the standard training strategy (B1).

| Student (teacher) | | Teacher | Student | KD [16] | OFD [14] | CRD [41] | SRRL [47] | Review [7] | DIST |
|---|---|---|---|---|---|---|---|---|---|
| ResNet-18 (ResNet-34) | Top-1 | 73.31 | 69.76 | 70.66 | 71.08 | 71.17 | 71.73 | 71.61 | **72.07** |
| | Top-5 | 91.42 | 89.08 | 89.88 | 90.07 | 90.13 | 90.60 | 90.51 | 90.42 |
| MobileNet (ResNet-50) | Top-1 | 76.16 | 70.13 | 70.68 | 71.25 | 71.37 | 72.49 | 72.56 | **73.24** |
| | Top-5 | 92.86 | 89.49 | 90.30 | 90.34 | 90.41 | 90.92 | 91.00 | 91.12 |

## 4 Experiments

### 4.1 Experimental settings

**Training strategies.** The training strategies of image classification task are summarized in Table 1. **CIFAR-100.** For fair comparisons, we use the same training strategies (referred to *A1* in Table 1) and pretrained models following CRD [41]. **ImageNet.** B1: for comparisons with previous KD methods, we train our baselines with the same simple training strategy as CRD [41]. B2: to validate the effectiveness of KD methods on modern training strategies, we follow EfficientNet [40] and design a training strategy B2, which can significantly improve the performance compared to B1. B3: the strategy B3 is used for training Swin-Transformers [27], and contains even more stronger data augmentations and regularization.

**Loss weights.** On CIFAR-100 and ImageNet, we set $\alpha = 1$, $\beta = 2$, and $\gamma = 2$ in Eq.(10). On object detection and semantic segmentation, these three factors are all equal to 1. For KD [16], we set $\alpha = 0.9$, $\beta = 1$ in Eq.(3), and use a default temperature $\tau = 4$. Specifically, instead of using $\tau = 1$ on ImageNet, we choose a larger temperature $\tau = 4$ on CIFAR-100, as it is easy to get overfit and the learned probabilistic distribution is sharp on CIFAR-100.

### 4.2 Image Classification

**Baseline results on ImageNet.** We first compare our method with prior works using the baseline settings. As shown in Table 2, our DIST significantly outperforms prior KD methods. Note that our method is only conducted on the outputs of models, and has a similar computational cost as KD [16]. Nevertheless, it even achieves better performance compared to those sophisticatedly-designed methods. For example, CRD [41] needs to preserve a memory bank for all 128-d features of ImageNet images, and produces additional 260M FLOPs of computation cost; SRRL [47] and Review [7] require additional convolutions for feature alignments. The implementation of DIST can be found in Appendix A.1, which is quite simple compared to these methods.

**Distillation from stronger teacher models.** As the stronger teachers come from larger model sizes and stronger strategies, we here first conduct experiments to compare our DIST with the vanilla KD on different scales (model sizes) of ResNets with baseline strategy B1. As shown in Table 3, when the teacher goes larger, the ResNet-18 students perform even worse than that with a medium-sized ResNet-50 teacher. Nevertheless, our DIST shows an upward trend with larger teachers, and the improvements compared to KD also become more significant, indicating that our DIST tackles better on the large discrepancy between the student and larger teacher.

**Distillation from stronger training strategies.** Recently, the performance of models on ImageNet has been significantly improved by the sophisticated training strategies and strong data augmentations

Table 3: **Performance of ResNet-18 and ResNet-34 on ImageNet with different sizes of teachers.**

| Student | Teacher | Top-1 ACC (%) | | | |
|---|---|---|---|---|---|
| | | student | teacher | KD | DIST |
| ResNet-18 | ResNet-34 | 69.76 | 73.31 | 71.21 | **72.07** (+0.86) |
| | ResNet-50 | | 76.13 | 71.35 | **72.12** (+0.77) |
| | ResNet-101 | | 77.37 | 71.09 | **72.08** (+0.99) |
| | ResNet-152 | | 78.31 | 71.12 | **72.24** (+1.12) |
| ResNet-34 | ResNet-50 | 73.31 | 76.13 | 74.73 | **75.06** (+0.33) |
| | ResNet-101 | | 77.37 | 74.89 | **75.36** (+0.47) |
| | ResNet-152 | | 78.31 | 74.87 | **75.42** (+0.55) |

Table 4: **Performance of students trained with strong strategies on ImageNet.** The *Swin-T* is trained with strategy B3 in Table 1, others are trained with B2. †: trained by [44]. ‡: Pretrained on ImageNet-22K.

| Teacher | Student | Top-1 ACC (%) | | | | | |
|---|---|---|---|---|---|---|---|
| | | teacher | student | KD [16] | RKD [30] | SRRL [47] | DIST |
| ResNet-50† | ResNet-18 | 80.1 | 73.4 | 72.6 | 72.9 | 71.2 | **74.5** |
| | ResNet-34 | | 76.8 | 77.2 | 76.6 | 76.7 | **77.8** |
| | MobileNetV2 | | 73.6 | 71.7 | 73.1 | 69.2 | **74.4** |
| | EfficientNet-B0 | | 78.0 | 77.4 | 77.5 | 77.3 | **78.6** |
| Swin-L‡ | ResNet-50 | 86.3 | 78.5 | 80.0 | 78.9 | 78.6 | **80.2** |
| | Swin-T | | 81.3 | 81.5 | 81.2 | 81.5 | **82.3** |

(*e.g.*, TIMM [44] achieves 80.4% accuracy on ResNet-50 while the baseline strategy B1 only obtains 76.1%). However, most of the KD methods still conduct experiments with simple training settings. It is seldomly investigated whether the KD methods are suitable to the advanced strategies. In this way, we conduct experiments with advanced training strategies and compare our method with vanilla KD, instance relation-based RKD [30], and SRRL [47].

We first train traditional CNNs with strong strategies, and also use a strong ResNet-50 with 80.1% accuracy trained by [44] as the teacher. As results shown in Table 4, on both similar architectures (ResNet-18, ResNet-34) and dissimilar architectures (MobileNetV2, EfficientNet-B0), our DIST can achieve the best performance. Note that RKD and SRRL can perform worse than training from scratch, especially when the students are small (ResNet-18 and MobileNet) or the architectures of teacher and student are fairly different (ResNet-50 and Swin-L), this might be because they focus on the intermediate features, which can be more challenging for the student to recover teacher's features compared to predictions.

Furthermore, we experiment on the recent state-of-the-art Swin-Transformer [27]. The results show that our DIST gains improvements on even more stronger models and strategies. For example, with Swin-L teacher, our method improves ResNet-50 and Swin-T by 1.7% and 1.0%, respectively.

**CIFAR-100.** The results on CIFAR-100 dataset in Table 5 show that, by distilling on the predicted logits, our method even outperforms those sophisticatedly-designed feature distillation methods.

## 4.3 Object Detection

We further investigate the effectiveness of DIST on downstream tasks. We conduct experiments on MS COCO object detection dataset [25], and simply leverage our DIST as an additional supervision on the final predictions of classes. Following [37, 52], we use the same standard training strategies and utilize Cascade Mask R-CNN [2] with ResNeXt-101 backbone as the teacher for two-stage student of Faster R-CNN [23] with ResNet-50 backbone; while for one-stage RetinaNet [24] with ResNet-50 backbone, the RetinaNet with ResNeXt-101 backbone is utilized as the teacher.

As shown in Table 6, our DIST achieves competitive results on COCO validation set. For comparisons, we train the vanilla KD under the same settings as our DIST, the results show that our DIST significantly outperforms vanilla KD by simply replacing the loss functions. Moreover, by combining

Table 5: **Evaluation results on CIFAR-100 dataset.** The upper and lower models denote teacher and student, respectively.

| Method | Same architecture style | | | Different architecture style | | |
| --- | --- | --- | --- | --- | --- | --- |
| | WRN-40-2 WRN-40-1 | ResNet-56 ResNet-20 | ResNet-32x4 ResNet-8x4 | ResNet-50 MobileNetV2 | ResNet-32x4 ShuffleNetV1 | ResNet-32x4 ShuffleNetV2 |
| Teacher | 75.61 | 72.34 | 79.42 | 79.34 | 79.42 | 79.42 |
| Student | 71.98 | 69.06 | 72.50 | 64.6 | 70.5 | 71.82 |
| *Feature-based methods* | | | | | | |
| FitNet [35] | 72.24±0.24 | 69.21±0.36 | 73.50±0.28 | 63.16±0.47 | 73.59±0.15 | 73.54±0.22 |
| VID [1] | 73.30±0.13 | 70.38±0.14 | 73.09±0.21 | 67.57±0.28 | 73.38±0.09 | 73.40±0.17 |
| RKD [30] | 72.22±0.20 | 69.61±0.06 | 71.90±0.11 | 64.43±0.42 | 72.28±0.39 | 73.21±0.28 |
| PKT [31] | 73.45±0.19 | 70.34±0.04 | 73.64±0.18 | 66.52±0.33 | 74.10±0.25 | 74.69±0.34 |
| CRD [41] | 74.14±0.22 | 71.16±0.17 | 75.51±0.18 | 69.11±0.28 | 75.11±0.32 | 75.65±0.10 |
| *Logits-based methods* | | | | | | |
| KD [16] | 73.54±0.20 | 70.66±0.24 | 73.33±0.25 | 67.35±0.32 | 74.07±0.19 | 74.45±0.27 |
| DIST | 74.73±0.24 | 71.75±0.30 | 76.31±0.19 | 68.66±0.23 | 76.34±0.18 | 77.35±0.25 |

Table 6: **Results on COCO validation set.** T: teacher; S: student. *: We implement KD using $\tau = 1$ and other settings are the same as DIST.

| Method | AP | AP$_{50}$ | AP$_{75}$ | AP$_S$ | AP$_M$ | AP$_L$ |
| --- | --- | --- | --- | --- | --- | --- |
| *Two-stage detectors* | | | | | | |
| T: Cascade Mask RCNN-X101 | 45.6 | 64.1 | 49.7 | 26.2 | 49.6 | 60.0 |
| S: Faster RCNN-R50 | 38.4 | 59.0 | 42.0 | 21.5 | 42.1 | 50.3 |
| KD [16]* | 39.7 | 61.2 | 43.0 | 23.2 | 43.3 | 51.7 |
| FKD [52] | 41.5 | 62.2 | 45.1 | 23.5 | 45.0 | 55.3 |
| CWD [37] | 41.7 | 62.0 | 45.5 | 23.3 | 45.5 | **55.5** |
| DIST | 40.4 | 61.7 | 43.8 | **23.9** | 44.6 | 52.6 |
| DIST + mimic | **41.8** | 62.4 | 45.6 | 23.4 | **46.1** | 55.0 |
| *One-stage detectors* | | | | | | |
| T: RetinaNet-X101 | 41.0 | 60.9 | 44.0 | 23.9 | 45.2 | 54.0 |
| S: RetinaNet-R50 | 37.4 | 56.7 | 39.6 | 20.0 | 40.7 | 49.7 |
| KD [16]* | 37.2 | 56.5 | 39.3 | 20.4 | 40.4 | 49.5 |
| FKD [52] | 39.6 | 58.8 | 42.1 | 22.7 | 43.3 | 52.5 |
| CWD [37] | **40.8** | **60.4** | **43.4** | 22.7 | **44.5** | **55.3** |
| DIST | 39.8 | 59.5 | 42.5 | 22.0 | 43.7 | 53.0 |
| DIST + mimic | 40.1 | 59.4 | 43.0 | **23.2** | 44.0 | 53.6 |

Table 7: **Results on Cityscapes val dataset.** All models are pretrained on ImageNet.

| Method | mIoU (%) |
| --- | --- |
| T: DeepLabV3-R101 | 78.07 |
| S: DeepLabV3-R18 | 74.21 |
| SKD [26] | 75.42 |
| IFVD [43] | 75.59 |
| CWD [37] | 75.55 |
| CIRKD [46] | 76.38 |
| DIST | **77.10** |
| S: PSPNet-R18 | 72.55 |
| SKD [26] | 73.29 |
| IFVD [43] | 73.71 |
| CWD [37] | 74.36 |
| CIRKD [46] | 74.73 |
| DIST | **76.31** |

DIST with mimic, which minimizes the mean square error between FPN features of teacher and student, we can even outperform the state-of-the-art KD methods designed for object detection.

## 4.4 Semantic Segmentation

We also perform experiments on semantic segmentation, a challenging dense prediction task. Following [37, 43, 46], we train DeepLabV3 [6] and PSPNet [54] with ResNet-18 backbone on Cityscapes dataset, and adopt our DIST on the predictions of classification head using a teacher with ResNet-101 backbone of DeepLabV3. As the results summarized in Table 7, with only the supervision of class predictions, our DIST can significantly outperform existing knowledge distillation methods on semantic segmentation task. For example, our DIST outperforms recent state-of-the-art method CIRKD [46] by 1.58% on PSPNet-R18. This demonstrate our effectiveness on relation modeling.

## 4.5 Ablation studies

**Effects of inter-class and intra-class correlations.** This paper proposes two types of relations: inter-class and intra-class relations. To validate the effectiveness of each relation, we conduct experiments to train students with these relations separately. The results on Table 8 verify that, both inter-class and intra-class relations can outperform the vanilla KD; also, the performance could be further boosted by combining them together.

Table 8: **Ablation of inter-class and intra-class relations on ImageNet.** The student and teacher models are ResNet-18 and ResNet-34, respectively.

| Method | Inter | Intra | ACC (%) |
|---|---|---|---|
| KD | - | - | 71.21 |
| DIST (KL div.) | ✘ | ✔ | 70.61 |
| DIST (KL div.) | ✔ | ✔ | 71.62 |
| DIST | ✔ | ✘ | 71.63 |
| DIST | ✘ | ✔ | 71.55 |
| DIST | ✔ | ✔ | **72.07** |

**Effect of intra-class relation in vanilla KD.** To investigate the effectiveness of intra-class relation in vanilla KD, we adopt experiments to train our DIST using KL divergence as the relation metric, denoted as *DIST (KL div.)*[3]. As the results summarized in Table 8, adding intra-class relation in the vanilla KD can also improve the performance (from 71.21% to 71.62%). However, when the student is trained with intra-class relation only, the improvement of using KL divergence is less significant than using Pearson correlation (70.61% vs. 71.55%), since the means and variances of intra-class distributions could be varied.

**Effect of training students with KD loss only.** Training student with only the KD loss can better reflect the distillation ability and the information richness of supervision signals. As results in Table 9 show that, when the student is trained with only the KD loss, our DIST significantly outperforms the vanilla KD. Without using the ground-truth labels, it can even outperform the standalone training accuracy, which indicates the effectiveness of our DIST in distilling those truly-beneficial relations.

Table 9: **Comparisons of training KD with or without the classification loss on ImageNet.** The student and teacher models are ResNet-18 and ResNet-34, respectively. The original accuracy of ResNet-18 without KD is 69.76%.

| Method | w/ cls. loss | w/o cls. loss |
|---|---|---|
| KD | 71.21 | 68.12 |
| DIST | **72.07** | **70.65** |

More ablation studies can be found in Section A.3.

## 5 Conclusion

This paper presents a new knowledge distillation (KD) method named DIST to implement better distillation from a stronger teacher. We empirically study the catastrophic discrepancy problem between the student and a stronger teacher, and propose a relation-based loss to relax the exact match of KL divergence in a linear sense. Our method DIST is simple yet effective in handling strong teachers. Extensive experiments show our superiority in various benchmark tasks. For example, DIST even outperforms state-of-the-art KD methods designed specifically for object detection and semantic segmentation.

## Acknowledgements

This work was supported in part by the Australian Research Council under Project DP210101859 and the University of Sydney Research Accelerator (SOAR) Prize.

---

[3]Specifically, the vanilla KD is the same as DIST (KL div.) with inter-class relation only.

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
