# A  Appendix

## A.1  Implementation of DIST

This section presents the implementation code of DIST, as shown in Figure 4. With only the output logits of student and teacher models, our DIST leverages quite simple inputs and computations, and is easy to implement.

```python
import torch.nn as nn

def cosine_similarity(a, b, eps=1e-8):
    return (a * b).sum(1) / (a.norm(dim=1) * b.norm(dim=1) + eps)

def pearson_correlation(a, b, eps=1e-8):
    return cosine_similarity(a - a.mean(1).unsqueeze(1), b - b.mean(1).unsqueeze(1), eps)

def inter_class_relation(y_s, y_t):
    return 1 - pearson_correlation(y_s, y_t).mean()

def intra_class_relation(y_s, y_t):
    return inter_class_relation(y_s.transpose(0, 1), y_t.transpose(0, 1))

class DIST(nn.Module):
    def __init__(self, beta, gamma):
        super(DIST, self).__init__()
        self.beta = beta
        self.gamma = gamma

    def forward(self, z_s, z_t):
        y_s = z_s.softmax(dim=1)
        y_t = z_t.softmax(dim=1)
        inter_loss = inter_class_relation(y_s, y_t)
        intra_loss = intra_class_relation(y_s, y_t)
        kd_loss = self.beta * inter_loss + self.gamma * intra_loss
        return kd_loss
```

Figure 4: The PyTorch [32] implementation of DIST.

## A.2  Related work

There exist some KD methods [4, 30, 34] which transfer the relations between instances from teacher to student (see Figure 3 for illustration). The purpose of these methods is to learn the similarity relationships between instances from teacher, *e.g.*, the semantic spaces of instances with the same class should be close, while the instances of different classes should have larger distances. To distill this instance relation, CCKD [34] proposes to measure the correlation between each instance pairwisely, then minimizes the difference of correlation scores between teacher and student; RKD [30] formulates the instance relation via the distance and angle between each pair; 4 introduces a locality preserving loss to embed the teacher's instance relation into student.

Nevertheless, in this paper, we focus on the probabilistic prediction instead of intermediate features, and leverage the relation on a new perspective to relax and replace the KL divergence in almost all KD methods, which help us to achieve better performance especially when the student is trained with a stronger teacher. Besides, different from previous works which measure relations between instances of teacher and student individually, we propose inter-class relation and intra-class relation simultaneously to better distill from a teacher.

## A.3  More ablation studies

**Correlations between teacher and student.** To validate the effectiveness of our correlation-based loss, we measure the correlations between teacher and student models, where the student models are trained by plain classification loss, KD, and our DIST. We choose commonly used Pearson correlation coefficient, Spearman's [10] and Kendall's Tau [20] rank correlation coefficients as the metrics of correlation. As summarized in Figure 5, our DIST obtains higher inter-class and intra-class correlations compared to baselines.

**Using cosine similarity in DIST.** In our method, the relation matching can be any function with the same form of Eq.(4). We simply adopt a commonly used Pearson correlation as our relation metric in DIST. Here we conduct experiments to investigate the efficacy of our method with cosine similarity.

Both cosine similarity and Pearson correlation coefficient can evaluate the relations between teacher and student. Compared to the scale invariance in cosine similarity, the Pearson correlation has an

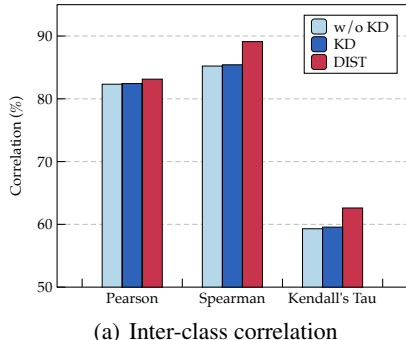
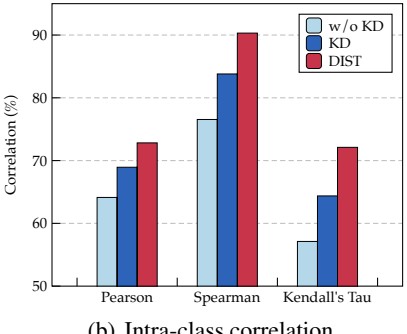

(a) Inter-class correlation        (b) Intra-class correlation

Figure 5: **Correlations between ResNet-18 student and ResNet-34 teacher.** We train the methods on ImageNet with B1 strategy.

Table 10: **Ablation of cosine similarity and Pearson correlation in DIST.** We train the student ResNet-18 and teacher ResNet-34 on ImageNet with or without label smoothing (LS).

| Method | w/o LS | w/ LS |
|---|---|---|
| Teacher | 73.31 | 73.78 |
| KD ($\tau = 4$) | 71.21 | 70.71 |
| KD ($\tau = 1$) | 71.49 | 71.37 |
| DIST (cosine) | 71.79 | 71.63 |
| DIST (Pearson) | **72.07** | **72.18** |

additional shift-invariance by centering the vectors first (see Eq.(7)), and it could be more robust to the distribution changes. We conduct experiments to compare these two metrics in our DIST and train the models with or without label smoothing. Since recent studies [3, 36] state that KD with high temperatures is incompatible with label smoothing, we also train the models with KD ($\tau = 1$). As shown in Table 10, adopting DIST with Pearson correlation achieves higher accuracies compared to KD and DIST with cosine similarity, especially when the teacher and student are trained with label smoothing (the predicted probabilistic distributions would be shifted by it). As a result, the scale-and-shift-invariant Pearson correlation may be a better metric for measuring relations in DIST.

## A.4 Comparisons of training speed

We compare the training speed of our DIST with vanilla KD [16], RKD [30], CRD [41], and SRRL [47], as summarized in Table 11. Our DIST has almost the same highest training speed as the vanilla KD, outperforming other feature-based KD methods.

Table 11: Average training speed (batches / second) of training ResNet-18 student with ResNet-34 teacher on ImageNet using strategy B1. The speed is tested based on our implementations on 8 NVIDIA V100 GPUs.

| KD | RKD | SRRL | CRD | DIST |
|---|---|---|---|---|
| [16] | [30] | [47] | [41] | |
| 14.28 | 11.11 | 12.98 | 8.33 | 14.19 |

## A.5 Landscapes of matching functions

As discussed in our main text, the matching functions such as KL divergence and MSE are used to match the outputs between student and teacher in KD. In DIST, we propose to relax the match in KD with relational matching functions, *i.e.*, Pearson distance. Here we visualize the landscapes of KL divergence, MSE, cosine distance, and Pearson distance to better show our effectiveness on relaxing the constraints. As shown in Figure 6 and Figure 7, we randomly generate a 2-dimensional vector as the output logits of student and teacher, then we adjust the first element in the teacher and

student logits (namely, $Z_1^{(s)}$ and $Z_1^{(t)}$) independently and fix the remaining elements, to draw the loss or gradient landscapes w.r.t. $Z_1^{(s)}$ and $Z_1^{(t)}$. Specifically, when $Z_1^{(s)} = Z_1^{(t)}$, the logits of student and teacher are identical, and the loss values of these functions should be zero. When we change the values of logits, the curves shows that how the losses are changed by it, *i.e.*, the sensitivity of loss to the distribution shifts. The landscapes of gradient in Figure 7 show that, the PCC in our DIST has sharper gradients curves in non-optimal regions and has the largest area on the region of optima. This indicates that our DIST can get optimized quickly with wider ranges of $Z_1^{(t)}$ and $Z_1^{(s)}$, thus it would have smaller conflict to the task loss.

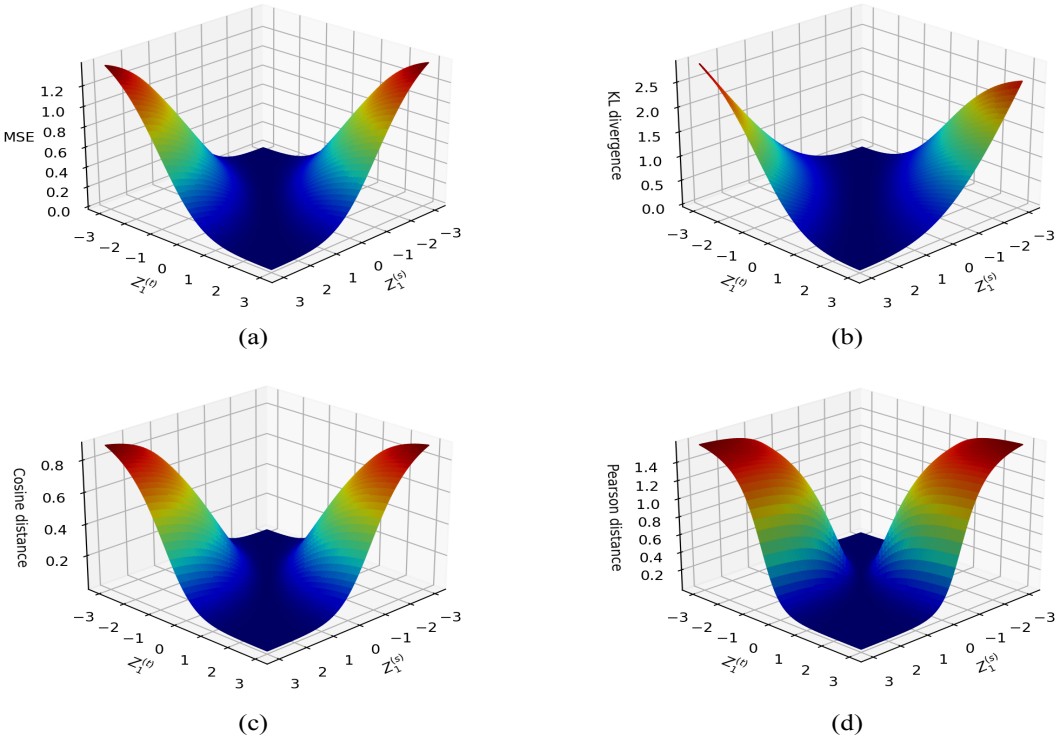

Figure 6: Visualization of loss landscapes. (a) MSE. (b) KL divergence. (c) Cosine distance. (d) Pearson distance.

## A.6 Discussion

**Limitations.** DIST can improve multi-class classification consistently in various tasks such as classification, object detection, and semantic segmentation. However, it would be less-effective on binary classification task, as the task only contains two classes and the information in inter-class relation is limited.

**Societal impacts.** Investigating the efficacy of the proposed method would consume considerable computing resources. These efforts can contribute to increased carbon emissions, which could raise environmental concerns. However, the proposed knowledge distillation method can improve the performance of light-weight compact models, where replacing the heavy models with light models in production could save more energy consumption, and it is necessary to validate the efficacy of DIST adequately.

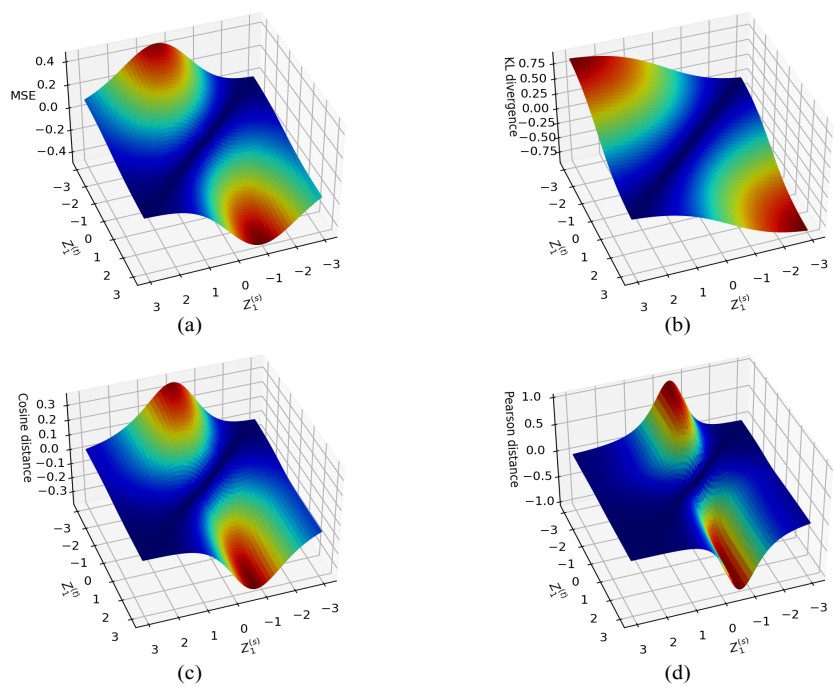

Figure 7: Visualization of gradient landscapes.