# OpenReview forum: "Knowledge Distillation from A Stronger Teacher"
_NeurIPS.cc/2022/Conference — NeurIPS 2022 Accept_

### Official Review · Reviewer_FMUS · 2022-06-30

**Rating:** 5
**Confidence:** 5
**Soundness:** 2 fair
**Presentation:** 3 good
**Contribution:** 2 fair

**Summary:**

This paper proposes a new knowledge distillation method that can use a stronger teacher to make a better student. It proposes a correlation-based loss at the inter-class level and intra- class level. The distance between the teacher predictions and the student predictions is computed using the Pearson correlation coefficient instead of KL divergence. Experiments show the effectiveness of the proposed method.

**Questions:**

See Weaknesses.

**Limitations:**

The authors have discussed the limitations and societal impacts of their works. The proposed method cannot fully address the binary classification tasks.

**Strengths And Weaknesses:**

Strengths:
1)	Existing KD methods perform worse when the teacher becomes stronger, while the proposed method can use a stronger teacher to train a better student.
2)	Experimental results verify the effectiveness.


Weaknesses:
1)	One of the key components is the matching metric, namely, the Pearson correlation coefficient (PCC). However, the assumption that PCC is a more relaxed constraint compared with KL divergence because of its invariance to scale and shift is not convincing enough. The constraint strength of a loss function is defined via its gradient distribution. For example, KL divergence and MSE loss have the same optimal solution while MSE loss is stricter than KL because of stricter punishment according to its gradient distribution. From this perspective, it is necessary to provide the gradient comparison between KL and PCC.
2)	The experiments are not sufficient enough.
2-1) There are limited types of teacher architectures.
2-2) Most compared methods are proposed before 2019 (see Tab. 5).
2-3) The compared methods are not sufficient in Tab. 3 and 4.
2-4) The overall performance comparisons are only conducted on the small-scale dataset (i.e., CIFAR100). Large datasets (e.g., ImageNet) should also be evaluated.
2-5) The performance improvement compared with SOTAs is marginal (see Tab. 5). Some students only have a 0.06% gain compared with CRD.
3)	There are some typos and some improper presentations. The texts of the figure are too small, especially the texts in Fig.2. Some typos, such as “on each classes” in the caption of Fig. 3, should be corrected.

---

> ### Author Response · Authors · 2022-08-02
> **Analysis Pearson correlation coefficient on the gradient distribution perspective**
>
> Thanks for your valuable comments and efforts in reviewing our paper. We address your comments and questions in the following content.
>
> **Q1: The assumption that PCC is a more relaxed constraint compared with KL divergence because of its invariance to scale and shift is not convincing enough.**
>
> **A1:** Thanks for your detailed and helpful suggestions. The derivative of PCC w.r.t the logits is complex and cannot be easily compared to MSE and KL divergence through their functions. Therefore, we compare the gradient distributions of MSE, KL divergence, cosine distance, and Pearson distance through the visualization of a toy experiment. Concretely, we visualize the loss and gradient landscapes to show our superiority in relaxing the constraints (see A.5 in revision for details). The landscapes of gradient show that, the PCC in our DIST is less sensitive to the distribution shifts of logits; it has sharper gradient curves in non-optimal regions and has the largest area in the region of optima. This indicates that our DIST can get optimized quickly with wider ranges of $Z^{(t)}_1$ and $Z^{(s)}_1$; therefore, it is more relaxed and would have more consistent optimization goal with the supervised CE loss.

---

> ### Author Response · Authors · 2022-08-02
> **Explanations about our experiments**
>
> **Q2-1: There are limited types of teacher architectures.**
>
> **A2-1:** Besides using ResNets and Swin-transformers as teachers, we have added experiments on more teacher architectures (recently proposed ConvNeXt-L and RegNetY-32GF) on ImageNet, and the results are summarized in the following table. Our DIST also achieves significant improvements compared to the KD baseline.
>
> |Teacher|Student|KD ($\tau=4$)|DIST|
> |:--:|:--:|:--:|:--:|
> |ConvNeXt-L (86.6)|EfficientNet-B0 (78.0)|77.4|78.7|
> |RegNetY-32GF (80.8)|EfficientNet-B0 (78.0)|77.2|78.6|
>
> **Q2-2: Most compared methods are proposed before 2019 in Table 5.**
>
> **A2-2:** Thanks for your comments. We have added more recent methods in Table 5 in our revision.
>
> **Q2-3: The compared methods are not sufficient in Table 3 and 4.**
>
> **A2-3:** Since DIST is the first work that proposes to perform KD with much stronger teachers and advanced training strategies, no previous work directly has the results on our settings (especially for ViTs with stronger training strategies), and we need to implement the previous methods for comparison. However, due to the limitation of our computation resources, we are unable to implement a large number of methods for comparisons on large scale ImageNet dataset. Therefore, we also report our performance on the baseline settings in Table 2, where most of previous methods can be fairly compared. As summarized in Table 2, our DIST can significantly outperforms the previous methods.
>
> **Q2-4: The overall performance comparisons are only conducted on small-scale dataset (CIFAR-100), large datasets (e.g., ImageNet) should also be evaluated.**
>
> **A2-4:** We want to clarify that we have sufficiently evaluated our DIST and compared it with other KD methods through numerous experiments on ImageNet. The most commonly-used settings on ImageNet are in Table 2, which are used as a typical benchmark of KD methods, and comparing our DIST with recent state-of-the-art methods in Table 2 is sufficient to validate our efficacy.
>
> **Q2-5: The performance improvement compared with SOTAs is marginal on CIFAR-100.**
>
> **A2-5:** It is well-known that CIFAR-100 dataset is easy to overfit, with the highly confidently predicted soft targets, response-based methods (e.g., KD) are difficult to be competitive to those feature-based methods, as the features contain more information on CIFAR-100. Nevertheless, our response-based method DIST can still improve KD and outperform most feature-based methods. Our superiority is more significant on large-scale challenging ImageNet. For example, for ResNet-18 student, DIST outperforms KD by 1.4% in Table 2 and 1.9% using stronger strategy in Table 4.
>
> **Q3: There are some typos and some improper presentations.**
>
> **A3:** Thanks for your detailed comments. We have corrected them in our revision.

---

> ### Author Response · Authors · 2022-08-07
> **Further discussion to Reviewer FMUS**
>
> Dear Reviewer FMUS,
>
> We sincerely thank you for your efforts in reviewing our paper. We have provided corresponding responses and results, which we believe have covered your concerns. We hope to further discuss with you whether your concerns have been addresses or not. Please let us know if you still have any unclear part of our work.
>
> Best,
> Authors

---

### Official Review · Reviewer_kJN1 · 2022-07-11

**Rating:** 7
**Confidence:** 4
**Soundness:** 3 good
**Presentation:** 4 excellent
**Contribution:** 3 good

**Summary:**

This paper investigates the topic of learning from a stronger teacher in KD. The authors show that using KL divergence in KD may not perform well when distilling from stronger teachers, and propose a new KD method called DIST to only preserve the relations between the teacher and student outputs instead of exactly matching in KL divergence. Numerical results on different tasks are provided to show the superiority of DIST on baseline and stronger teacher settings.

**Questions:**

1. The authors adopt Pearson correlation as the "relation" metric in DIST, will the performance also be significant when having other relation metrics?

**Limitations:**

Yes, the authors adequately discussed the limitations of this paper.

**Strengths And Weaknesses:**

Strengths:
1. The problem of learning from a stronger teacher is interesting and worth to investigate. Many previous works have shown that the KD performs poorly when the capacity gap between teacher and student is large. This paper extends this topic to "stronger teacher", where the "stronger" denotes larger capacity or stronger training strategy, and provides a unified solution.

2. The proposed method is simple and effective. The authors empirically find that the discrepancy between the student and a stronger teacher becomes larger, and therefore adopt correlation coefficient to alleviate this discrepancy. The method is intuitively sound and well supported by the experiments.

3. The improvements compared to KD is significant according to the numerous experiments on various tasks.

Weaknesses:
1. DIST is less effective on larger student networks. For example, in Table 4, DIST achieves 1.9% improvement on ResNet18 compared to KD, but only improves ResNet34 and ResNet50 by 0.6% and 0.2%.

2. The performance of prediction-based distillation method is limited on dense prediction tasks (e.g., detection) compared to feature-based methods, as the feature contains more localization information.

---

> ### Author Response · Authors · 2022-08-02
> **Response to Reviewer kJN1**
>
> Thanks for your positive comments. Our responses to your comments and questions are as follows.
>
> **Q1: DIST is less effective on larger student networks.**
>
> **A1:** Our DIST achieves significant improvements on small students (e.g., DIST improves KD by 1.9% (ResNet-18) and 2.7% (MobileNetV2) in Table 4). While on large student models such as ResNet-50, we think the students might be strong enough to learn exactly from the teacher, and thus the vanilla KD could also obtain competitive performance. However, our DIST can also gain improvements, which is actually considered to be significant for large models, e.g., 81.5% (KD) vs. 82.3% (DIST) on Swin-T.
>
> **Q2: The performance of prediction-based distillation method is limited on dense prediction tasks compared to feature-based methods.**
>
> **A2:** Thanks for your useful comments. We agree that the feature is more effective on object detection task as it contains both recognition and localization information. Nevertheless, our DIST can be combined with feature-based methods to enhance the KD performance further. As shown in Table 6, our DIST combined with mimic strategy could achieve competitive performance compared to those methods specially designed for object detection, and even outperforms the state-of-the-art methods on Faster RCNN-R50 student.
>
> **Q3: The authors adopt Pearson correlation as the "relation" metric in DIST, will the performance also be significant when having other relation metrics?**
>
> **A3:** Thanks for your useful suggestions. We have also adopted another cosine similarity based match in our experiments (see Table 10 in our paper), The results show that, cosine similarity can also enjoy an obvious improvements over the KD baseline. We believe there are more types of relations such as non-linear correlations can also benefit the performance.

---

> > ### Comment · Reviewer_kJN1 · 2022-08-09
> > **Thanks for the response.**
> >
> > Thanks for the response. My concerns have been well addressed. This is a novel and effective method, which may inspire the community.

---

### Official Review · Reviewer_xqDf · 2022-07-12

**Rating:** 7
**Confidence:** 4
**Soundness:** 4 excellent
**Presentation:** 4 excellent
**Contribution:** 4 excellent

**Summary:**

This paper raise a overlooked question in knowledge distillation: how to distill from a strong teacher, i.e., how to conduct knowledge distillation when the discrepancy between the teacher and student models is large. This paper conducted an empirical study and found that existing KD methods may fail when distilling from a strong teacher. This paper further proposed to use the Pearson correlation coefficient as a new match manner to replace the KL divergence.

**Questions:**

Please see weaknesses.

**Limitations:**

The authors have addressed the limitations and potential negative societal impact of their work.

**Strengths And Weaknesses:**

Strengths:

\+ The proposed research question, how to distill from a strong teacher, is important but overlooked problem for knowledge distillation, which has a great potential in real-world applications. The research problem is novel and practical.

\+ This paper conducted a comprehensive empirical study on how previous KD methods fail when the discrepancy between the teacher and student models is large.

\+ This paper proposed to use the Pearson correlation coefficient as a new match manner to replace the KL divergence. The proposed method is simple, efficient, and practical.

\+ The paper conducted extensive experiments on various benchmark datasets, including image classification, object detection, and semantic segmentation.

\+ The paper is well written and organized.

Weaknesses:

\- The compared methods in Table 5 are not state-of-the-art. Some recent works (see References) are not compared.

\- The font size in Figures 1 and 2 is too small. It would be great if these two figures can be reorganized in the future version.

References:

[R1] Knowledge Distillation meets Self-supervision, ECCV 2020.
[R2] Densely Guided Knowledge Distillation using Multiple Teacher Assistants, ICCV 2021.

---

> ### Author Response · Authors · 2022-08-02
> **Response to Reviewer xqDF**
>
> Thanks for your efforts and positive comments. We respond to your comments and questions as follows.
>
> **Q1: The compared methods in Table 5 (CIFAR-100) are not state-of-the-art.**
>
> **A1:** Thanks for your comments. We have added these recent state-of-the-art methods in our revision. Note that current state-of-the-art methods on CIFAR-100 dataset are usually feature-based methods, as it is acknowledged that the feature contains more information on CIFAR-100. However, our logits-based method still outperforms a variety of feature-based methods and significantly outperforms the vanilla KD.
>
> **Q2: The font size in Figure 1 and 2 is too small.**
>
> **A2:** Thanks for your valuable suggestions. We have modified the font size of these two figures in our revision, and they are now more friendly to read.

---

### Official Review · Reviewer_RvKW · 2022-07-14

**Rating:** 5
**Confidence:** 2
**Soundness:** 3 good
**Presentation:** 2 fair
**Contribution:** 3 good

**Summary:**

The authors come up with a loss for KD that encourages not only the correlation of relative differences in scores of the class predictions for an image, but also the correlation of the relative differences of scores of the images for a class. They claim that this sort of loss improves KD from strong teacher models and they show improvements on ImageNet, COCO, Cityscapes etc.

**Questions:**

Some intuition behind why their row and column wise correlation based loss works better to distill knowledge would be nice, It was a bit hard to follow the dog, cat, airplane example.

**Limitations:**

Some qualitative examples and quantitative analysis on what fails to be distilled from the strong teachers would also be nice to have.

**Strengths And Weaknesses:**

Strengths:
- They show improvements on various benchmarks.
- Their proposed loss is straight forward to understand.

Weaknesses/Concerns:
- It is not clear to me how their described intuition in lines 142 to 148 relate to the column-wise loss where try to correlate the relative difference in scores per class.
- Some relevant literature references missing: https://arxiv.org/abs/1910.01348, https://arxiv.org/abs/1902.03393
- The title seems overly generic and doesn't describe the gist of the paper. KD is obviously done from a stronger teacher, that's not novel. The main novelty is the row and column wise loss.
- The teachers are also not that strong by current standards. They use a Resnet 18 and 50. Nowadays we have ViT's for image recongnition and other stronger models than resnet 18.

---

> ### Author Response · Authors · 2022-08-02
> **Response to Reviewer RvKW**
>
> Thank you for your positive comments and efforts in reviewing our paper. The responses to your comments and questions are as follows.
>
> **Q1: The intuition of the column-wise loss (intra-class relation).**
>
> **A1:** We have reworded the explanation of the intra-class relation in our revision (lines 143 to 149): _"Besides the inter-class relation, where we transfer the relation of different classes for each instance, the prediction scores of multiple instances for each class are also informative and useful. These scores indicate the similarities of multiple instances in the same class. For instance, suppose we have three images containing ‘cat’, ‘dog’, and ‘plane’, respectively, and they have three prediction scores on the ‘cat’ class, denoted as $e$, $f$, and $g$. Generally, the picture ‘cat’ should have the largest score to the ‘cat’ class, while the ‘plane’ should have the smallest score since it is inanimate. This relation of ‘$e>f>g$’ could also be benefitial and then transferred to the student."_
>
> **Q2: Some references were missing.**
>
> **A2:** Thanks for your advice, we have included these references in our revision (see lines 36 to 40).
>
> **Q3: The title seems overly generic and doesn’t describe the gist of the paper.**
>
> **A3:** Thanks for your helpful comments. We want to clarify that our paper's major motivation is to achieve better distillation from stronger teachers. To the best of our knowledge, this is the first attempt to unify the "KD with larger teacher models" and "KD with stronger strategies" topics into one unified topic: KD from a stronger teacher. We investigate the effects of stronger teachers from the distribution discrepancy perspective and propose to ease the KD objective by using relational matching principle. Finally, we leverage DIST as a typical solution to our intuition.
>
> **Q4: The ResNet teachers are not that strong by current standards.**
>
> **A4:** Thanks for your comments. In our paper, we actually have adopted Swin-Transformer (a typical variant of ViTs) as the teacher and student (see Table 4), where the Swin-L teacher achieves a fairly competing accuracy of 86.3% on ImageNet. We also included experiments for various types of teachers (recently proposed ConvNeXt-L and RegNetY-32GF) on ImageNet, and the results are summarized in the following table, which show that our DIST also achieves significant improvements compared to the KD baseline.
>
> |Teacher|Student|KD ($\tau=4$)|DIST|
> |:--:|:--:|:--:|:--:|
> |ConvNeXt-L (86.6)|EfficientNet-B0 (78.0)|77.6|78.7|
> |RegNetY-32GF (80.8)|EfficientNet-B0 (78.0)|77.4|78.6|
>
> **Q5: Some intuition behind why their row and column wise correlation based loss works better to distill knowledge would be nice.**
>
> **A5:** For the row-wise correlation based loss, our aim is to relax the point-to-point match in KL divergence, so that the KD loss would be less sensitive to the distribution shift of teacher models, and have more consistent optimization goal with the supervised cross-entropy loss. For the column-wise correlation based loss, we want to capture the relation from another dimension, i.e., the relations of one class to multiple instances, this could help our DIST distill more information from the teacher.

---

> > ### Author Response · Authors · 2022-08-07
> > **Further discussion to Reviewer RvKW**
> >
> > Dear Reviewer RvKW,
> >
> > We sincerely thank you for your efforts in reviewing our paper. We have provided corresponding responses and results, which we believe have covered your concerns. We hope to further discuss with you whether your concerns have been addresses or not. Please let us know if you still have any unclear part of our work.
> >
> > Best,
> > Authors

---

### Author Response · Authors · 2022-08-02
**Response to all reviewers**

We thank all the reviewers for their valuable feedback. We have addressed the writing problems in our revision, and all changes are marked in red color. The detailed responses to the reviewers’ comments will be replied directly to each reviewer.

---

### Meta-Review · Area_Chair_UEYn · 2022-08-24

**Recommendation:** Accept
**Confidence:** Certain

**Metareview:**

The paper presents a new KD loss different from the widely used KL divergence for learning from strong teachers who have large gaps between students. The authors provide a comprehensive study and improve the challenging benchmarks. The contribution is significant to the KD community and AC recommends accept. Authors may want to carefully upgrade the paper with constructive comments for the camera-ready version.



**Award:**

No

---

### Decision · Program_Chairs · 2022-09-14

Accept